# Dbf4 Zn-Finger Motif Is Specifically Required for Stimulation of Ctf19-Activated Origins in *Saccharomyces cerevisiae*

**DOI:** 10.3390/genes13122202

**Published:** 2022-11-24

**Authors:** Meghan V. Petrie, Haiyang Zhang, Emily M. Arnold, Yan Gan, Oscar M. Aparicio

**Affiliations:** Molecular and Computational Biology Section, Department of Biological Sciences, University of Southern California, 1050 Childs Way, Los Angeles, CA 90089-2910, USA

**Keywords:** replication origins, Dbf4-dependent-kinase, chromosome domains, Fkh1, Ctf19

## Abstract

Eukaryotic genomes are replicated in spatiotemporal patterns that are stereotypical for individual genomes and developmental profiles. In the model system *Saccharomyces cerevisiae*, two primary mechanisms determine the preferential activation of replication origins during early S phase, thereby largely defining the consequent replication profiles of these cells. Both mechanisms are thought to act through specific recruitment of a rate-limiting initiation factor, Dbf4-dependent kinase (DDK), to a subset of licensed replication origins. Fkh1/2 is responsible for stimulation of most early-firing origins, except for centromere (CEN)-proximal origins that recruit DDK via the kinetochore protein Ctf19, which is required for their early firing. The C-terminus of Dbf4 has been implicated in its recruitment to origins via both the Fkh1/2 and Ctf19 mechanisms. Here, we show that the Zn-finger motif within the C-terminus is specifically required for Dbf4 recruitment to CENs to stimulate CEN-proximal/Ctf19-dependent origins, whereas stimulation of origins via the Fkh1/2 pathway remains largely intact. These findings re-open the question of exactly how Fkh1/2 and DDK act together to stimulate replication origin initiation.

## 1. Introduction

Entry into S phase and initiation of replication of eukaryotic chromosomes is regulated by multiple cell cycle kinases, including essential activators Cyclin-Dependent Kinase (CDK) and Dbf4-Dependent Kinase (DDK). These major regulators are characterized by a kinase subunit activated and targeted by a protein that exhibits cell cycle regulation of its expression and stability. Dbf4 is the cell cycle-regulated, activating subunit of the Cdc7 kinase that together comprise DDK, an essential DNA replication kinase (reviewed in [1] As such, DDK is a node of regulation linking checkpoint signaling and establishment of cohesion during the replication process [2]. DDK’s essential function in replication initiation is to phosphorylate N-terminal domains of Mcm4 and Mcm6 at licensed origins to relieve intrinsic repression of MCM complex helicase activity [3].

DDK is a rate-limiting activator of replication origins [4,5,6]. As a result of DDK and other initiation proteins being present in limiting quantities in relation to the number of licensed origins, only a subset of origins can initiate replication simultaneously at the beginning of S phase in response to DDK and CDK activities. Mechanisms that specifically target DDK to subsets of origins have been described, resulting in strongly preferential activities of these origins (reviewed in [7]. For example, Fkh1 and/or Fkh2 (Fkh1/2), are responsible for the preferential activation of most early-firing origins in the *S. cerevisiae* genome with the exception of CEN-proximal origins, which remain early firing in *fkh1∆ fkh2∆* cells [8]. CEN-proximal origins are instead dependent on Ctf19, a kinetochore protein, for their early efficient activities [2]. Through apparently independent mechanisms, Fkh1/2 or Ctf19 is thought to recruit Dbf4, and hence DDK, to the selected origins, resulting in their preferential early initiation [2,8,9,10]. Fkh1/2 binding sequences are intimately associated with Fkh1/2-activated origins [8,11,12,13], whereas the Ctf19-kinetochore complex appears to recruit Dbf4 to the kinetochore itself, resulting in stimulation of the nearest flanking origin(s), up to ~25 kb distant [2].

Despite the independent DDK-origin recruitment mechanisms, the C-terminal region of Dbf4 has been implicated as a shared feature, required for both [9]. The C-terminus of Dbf4 contains a Zn-finger motif (CCHH), which was characterized previously, and suggested to play a direct role in replication through interaction with Mcm2, and a role in replication stress response [14] A subsequent study strongly suggested that the C-terminal domain functions in replication of CEN-proximal origins, as simply appending one of several epitope-tags on the C-terminus is sufficient to specifically ablate early firing of CEN-proximal origins; this was further shown to phenocopy *ctf19∆* regarding CEN-proximal origin firing and physical interaction [2]. More recently, the C-terminus of Dbf4 was implicated in the Fkh1/2-dependent stimulation of origins, suggesting direct physical interaction between the C-terminal Zn-finger region and Fkh1 [9]. These findings prompted us to follow up and examine this proposed mechanism in greater depth.

In this work, we have carefully examined and compared the cellular growth and viability, the rate of genome duplication, and the specific effects on individual origins genome-wide in *DBF4* mutants lacking the 50-amino acid Zn-finger domain or bearing point mutations precisely targeting the Zn-finger motif. We compare these results with strains bearing mutations that eliminate one or both origin-stimulating pathways, Fkh1/2 or Ctf19. Our findings show that deletion of the entire C-terminal domain causes a severe defect in replication and viability. In contrast, mutation of the Zn finger motif has specific effects on targeting to CEN-proximal versus Fkh-activated origins. These results significantly revise the possible mechanisms involved in origins stimulation, particularly of Fkh-activated origins.

## 2. Materials and Methods

### 2.1. Plasmid and Yeast Strain Construction

Oligonucleotide primer sequences for plasmid and strain constructions are given in Table 1. Plasmid pMP35 was constructed for pop-in/pop-out of *dbf4-Zn** mutant alleles to create precise, unmarked replacements of *DBF4*. Primers ∆N-dbf4_F and ∆N-dbf4_R were used to PCR-amplify an ∆N-dbf4 fragment with homology tails to pRS406 [15]. The fragment and pRS406 were digested with *KpnI-HF* and *SacI-HF* and ligated (enzymes from New England BioLabs). Zinc finger mutations were created in pMP35 with Quick-Change Lightning Multi Site-directed mutagenesis kit (Agilent) using primers dbf4-Zn-aaHH and dbf4-Zn-CCHc to create plasmids pMP36 and pMP38, respectively. Desired sequence changes were confirmed by DNA sequencing. Plasmid pHY62 was constructed with primers TB_FLAG-swap and swap-FLAG-Dbf4 replacing the 6xHA epitope in *KpnI-NaeI*-digested plasmid pT1892 (from T. Tanaka) with 3xFLAG epitope from plasmid p2L-3FLAG-TRP1 (from T. Tsukiyama) using the In-Fusion HD Cloning Kit (Clontech).

Strain constructions were carried out by genetic crosses or lithium acetate transformations with DNA from linearized plasmids or PCR products generated with hybrid oligonucleotide primers having homology to target loci [16,17,18]; Genomic alterations were confirmed by PCR analysis and/or DNA sequence analysis as appropriate. Yeast strain genotypes are given in Table 2. All strains are congenic with W303 background, and most are derived from BrdU-incorporating strains CVy63 and CVy70, which are derived from SSy161 and SSy162, respectively [19]. Primers dbf4∆C_F and dbf4∆C_R were used to create a construct truncating *DBF4* with selection (*dbf4∆C::HIS3MX6*) and introduced into diploid of CVy63 x CVy70. The diploid was sporulated and spores dissected and germinated at 23 °C; haploid spores were genotyped yielding strains JOSHy1 and JOSHy2. JOSHy1 was transformed with KANMX module to replace HIS3MX module to create HYy176. Deletions of *FKH1*, *FKH2*, and replacement of *FKH2* with *fkh2-dsm* have been described previously [20]; HYy143 and OAy1123 were similarly constructed. HYy215, HYy217 and HYy218 were obtained as segregants from a cross of HYy143 and JOSHy2. *CTF19* was deleted using primers CTF19∆_F and CTF19∆_R and plasmid p2L-3FLAG-TRP1(*Kluyveromyces lactis*) as template for the selectable marker in diploid of HYy143 x JOSHy2; sporulation and dissection yielded strains HYy207, HYy210 and HYy211. MPy74 and MPy76 were constructed by transformation of CVy63 with pMP36 and pMP38 digested with *EcoNI*. Following selection for *URA3* and confirmation of proper integration (pop-in), non-selective growth allowed for isolation of 5-FOA-resistant clones (pop-out); clones were sequenced to confirm replacement of *DBF4* by *dbf4-Zn*1* or *dbf4-Zn*2*. MPy86 and MPy90 were derived from MPy74 and MPy76, respectively, through a cross with OAy1123. N-terminal 3xFLAG-tagging of *DBF4* to construct MPy125 was accomplished by transformation of MPy35 with *XhoI*-digested pHY62; construction was confirmed by PCR and immunoblotting. EAy1 and EAy2 are sister isolates from pop-in/pop-out of pMP36 into strain MPy125 to introduce dbf4-Zn*1; constructions were confirmed by sequencing.

### 2.2. Other Methods

Except as otherwise noted in Figure 1, cultures were grown at 23 °C and G1-synchronization was performed as described previously using 5 ng/mL α-factor for *bar1∆* strains, and hydroxyurea (Sigma) at 200mM [21]. DNA content analysis by flow cytometry (FACScan) has been described previously (Aparicio et al. 2004). Preparation of protein extracts, PAGE, and immunoblotting were performed essentially as described previously [22], with the following specifics: 10% gel, semi-dry transfer, blot was incubated with polyclonal anti-Dbf4 serum (from B. Stillman) (1:1000) overnight at 4 °C, and detected with anti-rabbit secondary (1:5000, Sigma GENA934). Quantitative BrdU-IP-seq (QBU) analysis was performed as described [21]. ChIP-seq was performed as described [23], using anti-FLAG M2 (Sigma F1804) at 1:200. QBU and ChIP-seq libraries were constructed using KAPA Hyper Prep Kit (KK8504). High-throughput DNA sequencing was performed by the USC Genome Core or Novogene. Sequencing data is available at GEO (GSE215190).

### 2.3. Computation and Statistics

All sequencing data were binned (50bp) and median-smoothed over a 1kb window. The list of origins is from OriDB “confirmed” set (n = 410) [24] The FKH-activated list is from [8], and the CEN-proximal origin list is from [20]. Matlab was used for generation of most data displays and analyses.

## 3. Results and Discussion

### 3.1. Dbf4∆C Is Defective in Essential Dbf4 Function(s) beyond Origin Targeting by Fkh1 and Ctf19

To investigate the function in DNA replication of Dbf4′s C-terminal Zn finger domain, with a focus on its targeting of origins, we examined previously described *DBF4* alleles either lacking the C-terminal 50-amino acids (*dbf4∆C*), or containing point mutations precisely targeting the Zn finger motif (*dbf4-AAHH* (Zn*1) and *dbf4-CCHC* (Zn*2)), which we will collectively refer to as *dbf4-Zn** [9,14] (Figure 1A). During construction of the *dbf4∆C* strain, we observed that it was temperature sensitive at 30 °C (Figure 1B), which had been previously noted for a 45 amino acid deletion at 37 °C [14]. In contrast, the *dbf4-Zn** strains are not temperature sensitive at 30 °C, though they exhibit some sensitivity to growth at 37 °C (Figure 1B, 1st panel). The overall poor growth plus temperature sensitivity of *dbf4∆C* cells suggests a substantial defect in Dbf4∆C’s essential function to stimulate individual replication origins. By comparison, elimination of the known Dbf4 origin-targeting pathways individually or in combination in *fkh1∆ fkh2-*dsm (equivalent to *fkh1∆ fkh2∆* for origin regulation but lacking additional pleiotropic defects [22]), *ctf19∆C*, and *fkh1∆ fkh2-dsm ctf19∆C* cells, respectively, does not phenocopy the temperature sensitivity (Figure 1B, 2nd panel). Moreover, we have recently discovered that *dbf4∆C fkh1∆* cells are inviable at the permissive temperature for *dbf4∆C* [20], reinforcing the idea that Dbf4∆C is defective in function(s) in addition to its recruitment via Fkh1, while also suggesting that recruitment to origins by Fkh1 ameliorates the defect(s) of Dbf4∆C. In contrast, deletion of *CTF19* is viable in combination with *dbf4∆C* and does not enhance the *dbf4∆C* phenotype at higher temperatures, suggesting Ctf19 is not supporting Dbf4∆C function (Figure 1B, 3rd panel). Additionally, in contrast to the lethality of *FKH1* deletion in *dbf4∆C* cells, *FKH1* deletion is viable in *dbf4-Zn** cells, clearly indicating that Dbf4∆C lacks function(s) retained in the specific *dbf4-Zn** alleles (Figure 1B, 4th panel). It is also notable that deletion of *FKH1* enhances the temperature sensitivity of *dbf4-Zn** alleles at 37 °C, suggesting that *FKH1* function remains important in the *dbf4-Zn** mutant cells.

### 3.2. Dbf4∆C Is Defective in Overall Rate of Genome Replication While dbf4-Zn* and ctf19∆ Are Not

To gain more insight into the anticipated replication defects, we examined bulk genome replication by flow cytometry in the *dbf4* mutant strains. Cells were synchronized in G1 phase and released into S phase. In *WT* cells, DNA content mostly doubled between 30 and 60 min (Figure 2, left panel), though it is not possible to precisely define beginning and endpoints of replication by this type of analysis. The *dbf4-Zn** mutants showed similar timing of progression through and completion of S phase (Figure 2, left panel). However, *dbf4∆C* cells exhibited substantial delay in progressing through and completing S phase (Figure 2, left panel), consistent with a more severe defect than the Zn* mutants. Cells lacking Ctf19 (*ctf19∆*) exhibited no defect in S phase progression, whereas cells lacking both *FKH1* and *FKH2* origin stimulation function (*fkh1∆ fkh2-dsm*) exhibited a modest delay in S phase completion (Figure 2, center panel), perhaps of slightly greater magnitude than previously reported [8]. This is not surprising given the large numbers of origins dependent on Fkh1/2 for full activity [8]. Additional elimination of the Ctf19-dependent targeting pathway (*ctf19∆ fkh1∆ fkh2-dsm*), however, did not appear to enhance the replication delay of *fkh1∆ fkh2-dsm* cells (Figure 2, center panel). Cells lacking Fkh1 (*fkh1∆*) exhibited no defect in S phase progression, whereas deletion of *FKH1* in *dbf4-Zn** mutants caused a replication delay (Figure 2, right panel), indicating that Fkh1 contributes to function of Dbf4-Zn* as inferred earlier from the cell growth analysis. The replication delay of *fkh1∆ dbf4-Zn** is slightly greater than that of cells lacking both targeting pathways (*ctf19∆ fkh1∆ fkh2-dsm*), suggesting that Dbf4-Zn* has defects beyond the Ctf19 and Fkh1/2 targeting pathways. The data also indicate that while elimination of the Ctf19 targeting pathway has little if any effect on the rate of bulk genome replication, the Fkh1/2 pathway is needed for normal replication rate and is critical to sustain replication in *dbf4-Zn** cells.

Given the sensitivity of the Dbf4 C-terminus to epitope tagging, we wondered whether the deletion of the C-terminal domain resulted in destabilization of Dbf4. To examine protein levels of the different mutants, we performed immunoblots of crude extracts from G1-synchronized cells using a native anti-Dbf4 antibody (Appendix A). Dbf4 and Dbf4-Zn* were readily identified near their predicted size (78kDa), and the band shifts upward in a strain expressing N-terminally epitope-tagged DBF4 (Appendix A). Dbf4 and Dbf4-Zn* were present in similar abundances; however, Dbf4∆C (~73kDa) was not clear above the background, suggesting a reduction in its steady-state level and/or stability (Appendix A). Similar results were reported by Jones et al., though the proteins were expressed from heterologous promoters and epitope tagged for detection [14]. A reduction in quantity of a protein considered to normally be present in limiting quantity would seemingly explain Dbf4∆C’s overall replication defects described thus far by acting as a general hypomorph, in addition to defects in possible specific functions such as targeting.

### 3.3. Dbf4-Zn* Mutations Specifically Eliminate Early Activation of CEN-Proximal Origins

To delve more deeply into the specific replication defects, we used quantitative BrdU incorporation (QBU) in G1-synchronized cells released into S phase in the presence of hydroxyurea (HU) to generate genome-wide, early S-phase origin firing profiles. QBU data from replicates were examined for correlation (Appendix A), averaged, scale-normalized, and plotted for comparisons between experimental strains. A representative plot chromosome XV in *WT* cells yielded the expected early S-phase replication profile showing robust peaks of QBU signal at early/efficient origins (e.g., Fkh-activated, a few of which are identified, and CEN-proximal) and smaller QBU peaks at later/less-efficient origins (Figure 3A). The data for *dbf4∆C* cells showed profound defects in BrdU incorporation at most origins, including CEN-proximal origins, while a few relatively active origins dominate the early replication landscape (Figure 3A). Two-dimensional scatter plots allow global comparison of the mutant data in comparison to *WT* (Figure 3B). In the *dbf4∆C* strain, CEN-proximal origins exhibit severely reduced replication, as previously reported, while most of the remaining active origins are Fkh-activated origins (Figure 3A,B). This result is inconsistent with the C-terminus of Dbf4 being required for Fkh1-mediated recruitment.

The *dbf4-Zn** mutant strains exhibited similar chromosome XV replication profiles to each other but quite distinct from *WT* and *dbf4∆C* (Figure 3A). The *dbf4-Zn** replication profiles generally show robust activation of a subset of Fkh-activated origins, along with notably weak activation of CEN-proximal origins, and reduced activation of many other origins (Figure 3A,B). These results are consistent with the C-terminal Zn-finger motif having a crucial function in stimulation of CEN-proximal origins. The results further suggest that the C-terminal Zn-finger domain is not required for stimulation of origins via Fkh1; indeed, a subset of Fkh-activated origins appeared to dominate the replication landscape.

To examine the distribution of origin firing amongst major origin categories, we generated distribution boxplots for 410 previously confirmed replication origins sequences, 95 Fkh-activated origins, and 32 CEN-proximal origins, defined as the closest origin on each side of each centromere (Figure 3C). These distribution plots reinforce the conclusion that *dbf4∆C* has a profound effect on firing of all origins, including CEN-proximal, but a smaller reduction in Fkh-activated origins (Figure 3C). The *dbf4-Zn** mutants show similar but less severe reductions as *dbf4∆C*; however, CEN-proximal origins remain more comparatively reduced than Fkh-activated origins (Figure 3C, Appendix A).

A typical feature sometimes distinguishing replication profiles is the peak width, which is inversely correlated with the overall level of origins firing, reflecting greater fork progression enabled by reduced numbers of total forks initiated [25]. In the *dbf4-Zn** mutants, the dominant peaks were substantially wider than in *WT* cells, while much less QBU signal originated from other loci (Figure 3A,B). This result suggests that the relative rate of origin firing has tilted further in favor of a smaller number of origins, most of which are Fkh-activated origins. Given that both *dbf4∆C* and *dbf4-Zn** mutant cells show preferential firing of Fkh-activated origins, and that deletion of *FKH1* is lethal in combination with *dbf4∆C*, we conclude that the C-terminal Zn finger of Dbf4 is required for its recruitment to CEN-proximal origins but much less so for its recruitment through Fkh1/2. Still, there appears to be a defect in Dbf4-Zn* recruitment to other origins lacking either the Cft19 or Fkh1/2 mechanisms. In support of this, the *dbf4-Zn** mutants show a more severe genome-wide replication defect than *ctf19∆* cells, which are like *WT* cells in QBU profiles except for strong reduction in CEN-proximal origin firing (Figure 3A–C). Direct comparison of the changes in firing levels relative to *WT* (∆QBU) in *ctf19∆* versus *dbf4-Zn** strains illustrates their similar, stronger effects on CEN-proximal versus Fkh-activated origins (Figure 3D). As noted above, the *dbf4-Zn** mutants also show a degree of temperature sensitivity not shown by *ctf19∆ fkh1∆ fkh2-dsm* cells, which lack both recruitment pathways and consequently show a highly disrupted pattern of origin initiation (Figure 3A–C). Still, these targeting-defective cells exhibit relatively robust origin firing densities compared with the *dbf4-Zn** mutant strains (Figure 3A–C). Given the sensitivity of the C-terminal domain of Dbf4 to structural perturbation, we think the likeliest explanation is that the Zn-finger point mutations partly disrupt an aspect of Dbf4 structure/function leading to a mild, generalized replication defect (perhaps interaction with Mcm2) in addition to the fully penetrant defect in CEN-proximal origin targeting, presumably via the Ctf19 pathway.

### 3.4. CEN-Proximal Origins Are Differentially Sensitive to Loss of Ctf19 or Dbf4-Zn*

An interesting feature we noted differentiating chromosome replication profiles was the extent to which *CTF19* deletion, as well as the *dbf4-Zn** alleles, affected one versus both CEN-flanking origins. For example, from the scatter plots, it is apparent that a few CEN-proximal origins are minimally affected by *CTF19* deletion or *dbf4-Zn** (Figure 3B). A plot of chromosome X shows an example where one of the two CEN-flanking origins remains robustly active in *ctf19∆* and *dbf4-Zn** cells (Appendix A). This origin, ARS1014, remains a rather robust origin, even in *dbf4∆C* cells, as well as several other mutant combinations (Appendix A). However, this origin is strongly diminished by elimination of both Fkh1 and Fkh2 (Appendix A). We also observe, as previously, that deletion of *FKH1* often elevates the activity of other origins, particularly CEN-proximal origins, likely reflecting competition for rate-limiting replication factors [8]. These observations suggest that some origins retain other features that influence their level of function, at least in the absence of the recognized, dominant Fkh1/2 and Ctf19 mechanisms. We also noticed that some origins greater than 25 kbp distal to a CEN showed diminished activity in *ctf19∆* cells (Appendix A), indicating a somewhat more extensive domain over which Ctf19 can stimulate origin firing than previously reported [2].

### 3.5. Dbf4-Zn* Retains Fkh1-Dependent Targeting to Fkh1-Activated Origins

To directly test the hypothesis that Dbf4-Zn* continues to be directly recruited to origins by Fkh1/2, we performed QBU in *dbf4-Zn** mutants lacking *FKH1*. Deletion of *FKH1* in *WT* cells resulted in significant reduction in QBU signal of Fkh1-activated origins (n = 35, origins sensitive to the loss of only *FKH1* in *WT* cells [8]), such as *ARS1509, ARS1513.5* and *ARS1529.5*, as well as other Fkh-activated origins (Figure 4A–C, Appendix A). Deletion of *FKH1* in *dbf4-Zn** cells also resulted in a significant decrease in QBU signal for Fkh1-activated origins (Figure 4A–C, Appendix A), supporting the conclusion that *dbf4-Zn** mutants continue to be recruited to origins via Fkh1. These results further support the conclusion that the Zn-finger motif of Dbf4 is specifically involved with targeting to CEN-proximal origins but depends on a different motif or domain for recruitment to origins via Fkh1.

### 3.6. Dbf4-Zn* Is Defective in Its Recruitment to CENs

To directly test whether the recruitment of Dbf4 to replication origins is affected by Dbf4-Zn*, we performed ChIP-seq of Dbf4 and Dbf4-Zn*1, each bearing a triple-FLAG epitope at the N-terminus. Previous ChIP analysis of N-terminally tagged Dbf4 detected enrichment at several replication origins, and strong enrichment at CENs [2]. Our analysis of Dbf4 yielded similar results with robust enrichment at CENs and less, but detectable enrichment at CEN-proximal origins (Figure 5). However, we detected little to no enrichment at most other replication origins (Figure 5). Dbf4-Zn*1 showed significantly reduced enrichment at CENs, as well as reduced enrichment at CEN-proximal origins although the difference at the origins was not significant (Figure 5). Minor enrichment of Dbf4-Zn*1 was detected at other replication origins; however, the levels were not significantly different than for Dbf4 (Figure 5). These results strongly support the function of the Zn-finger motif in Dbf4′s targeting to CENs and consequent stimulation of CEN-proximal origins. However, these data do not allow us to conclude whether the Zn-finger motif plays a role in recruitment to other replication origins such as Fkh-activated origins. Nevertheless, the QBU data indicate that the Zn-finger motif is dispensable for most Fkh-dependent origin targeting.

## 4. Perspective

In retrospect, it may not be surprising that the C-terminal domain of Dbf4 is not the critical feature for physical interaction with Fkh1. It is anticipated that Fkh1 interacts with a phospho-regulated partner through its FHA domain, which is expected to function as a phosphothreonine binding module. However, no threonine resides amongst the C-terminal 50 amino acids of Dbf4, undermining the notion that this domain mediates the critical interaction with Fkh1-FHA. Further work will be required to define the exact nature of the interaction between Fkh1 and Dbf4, and with licensed origins.

## Figures and Tables

**Figure 1 genes-13-02202-f001:**
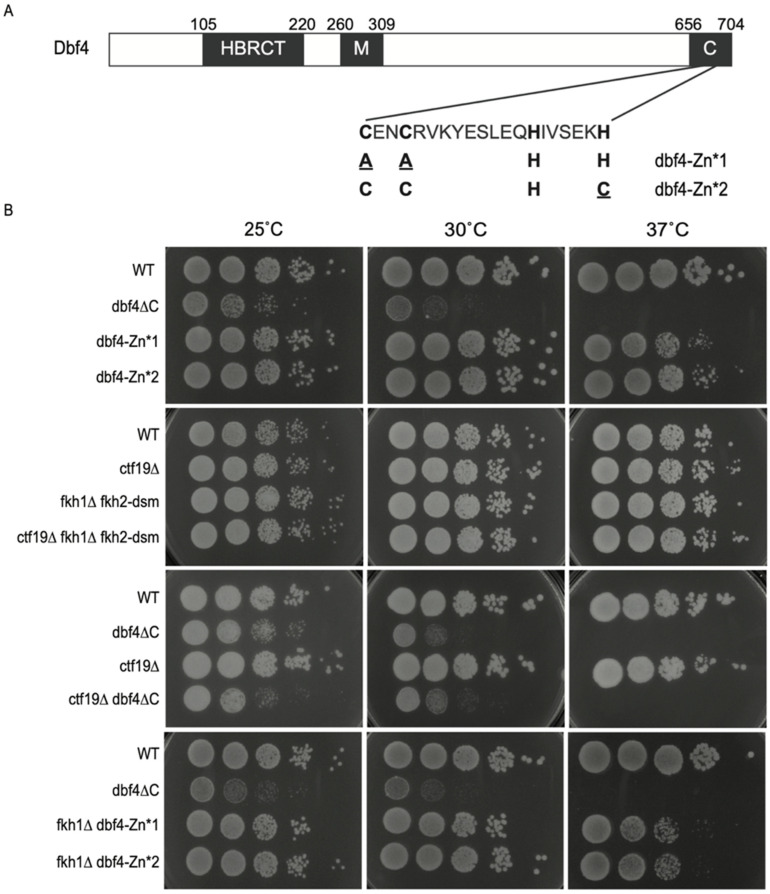
Dbf4∆C is defective in essential Dbf4 function(s). (**A**) Schematic representation of Dbf4 protein domains with sequence detail of the Zn-finger domain; amino acid changes in the Zn* alleles are indicated in lower case, red font. (**B**) Strains CVy63 (*WT*), HYy176 (*dbf4*ΔC), MPy74 (*dbf4-Zn*1*), MPy76 (*dbf4-Zn*2*), MPy86 (*dbf4-Zn*1 fkh1∆*), MPy90 (*dbf4-Zn*2 fkh1∆*), HYy210 (*cft19∆*), HYy211 (*cft19∆ dbf4∆C*), HYy207 (*cft19∆ fkh1*Δ *fkh2-dsm*), and HYy218 (*fkh1*Δ *fkh2-dsm*) were grown to mid-log phase, diluted, and plated onto rich media, then incubated at the indicated temperatures and imaged after 2–3 days.

**Figure 2 genes-13-02202-f002:**
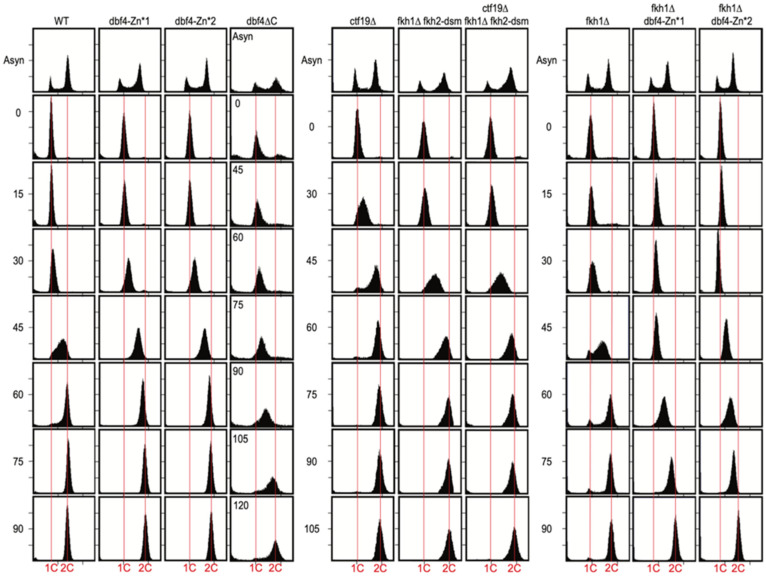
Dbf4∆C is defective in bulk genome replication, while Dbf4-Zn* mutants are not. Strains described in Figure legend 1 plus HYy217 (*fkh1∆)* and HYy215 (*dbf4∆C*—instead of HYy176) were synchronized in G1 phase and released into S phase for DNA content analysis. Note the different time intervals of the different panels, and the unique time intervals for *dbf4∆C* (inset). Vertical red lines indicate 1C (un-replicated) and 2C (replicated) chromosomal DNA contents, respectively.

**Figure 3 genes-13-02202-f003:**
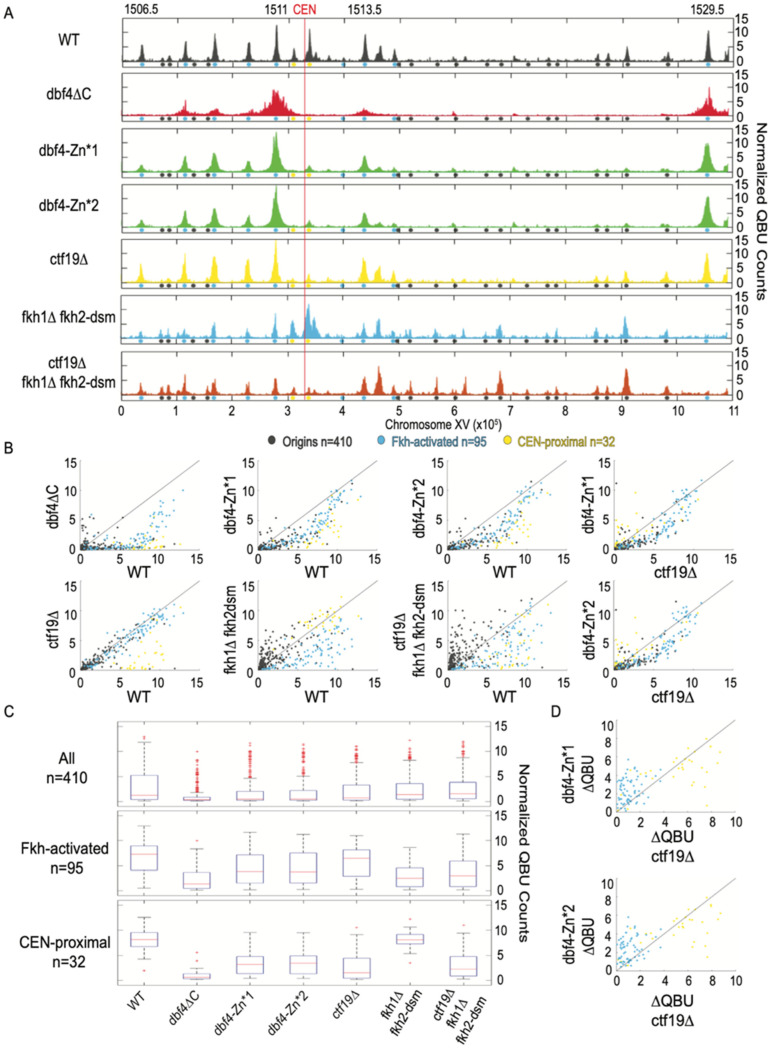
Dbf4-Zn* is defective in CEN-proximal origin firing. Strains described in Figure 1 and Figure 2 legends were synchronized in G1 phase, released into S phase with HU for 60 min (90 min for dbf4∆C), and harvested for QBU analysis. (**A**) QBU values averaged for two replicates and scale-normalized across strains are shown for representative chromosome XV; origins and origin sub-groups are indicated with color-coded circles below the x-axis. (**B**) Two-dimensional scatter plots comparing QBU signals averaged across 500 bp regions centered on 410 confirmed origins; origins and sub-groups are color-coded as indicated. (**C**) Boxplot distributions of averaged QBU counts across 500 bp regions aligned on origins of the indicated groups; the number of origins in each group is indicated within parentheses; results of statistical analyses are given in Appendix A. (**D**) Two-dimensional scatter plot comparing the differences in QBU values for ctf19∆ (WT-ctf19∆) versus dbf4-Zn* (WT-dbf4-Zn*) for Fkh-activated and CEN-proximal origins.

**Figure 4 genes-13-02202-f004:**
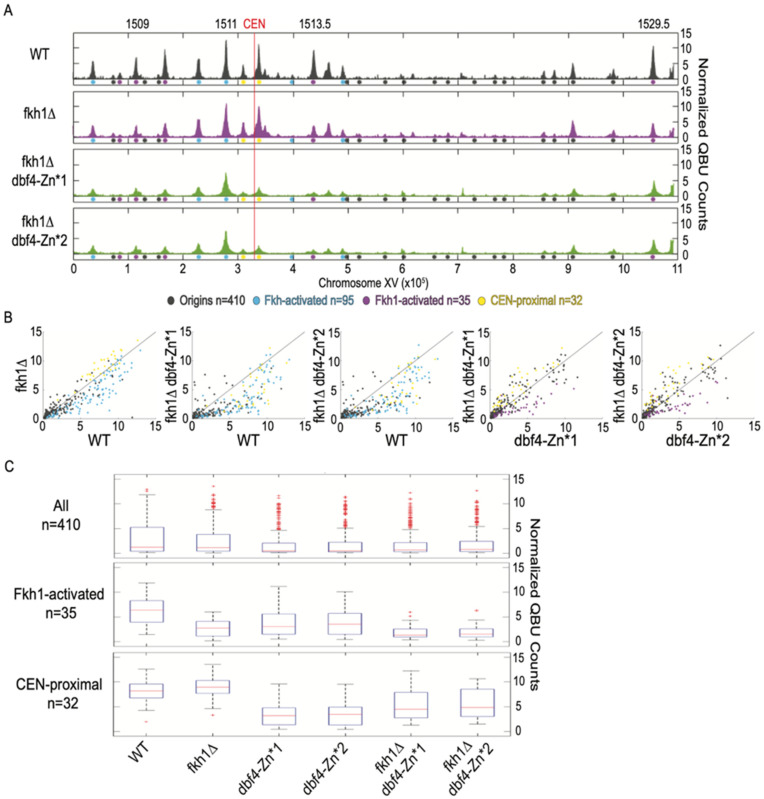
Dbf4-Zn* retains Fkh1 origin stimulation. Strains described in Figure 1 and Figure 2 legends were synchronized in G1 phase, released into S phase with HU for 60 min, and harvested for QBU analysis. (**A**) QBU values averaged for two replicates and scale-normalized across strains are shown for representative chromosome XV; origins and origin sub-groups are indicated with color-coded circles below the *x*-axis. (**B**) Two-dimensional scatter plots comparing QBU signals averaged across 500 bp regions centered on 410 confirmed origins; origins and sub-groups are color-coded as indicated. (**C**) Boxplot distributions of averaged QBU counts across 500 bp regions aligned on origins of the indicated groups; the number of origins in each group is indicated within parentheses; results of statistical analyses are given in Appendix A.

**Figure 5 genes-13-02202-f005:**
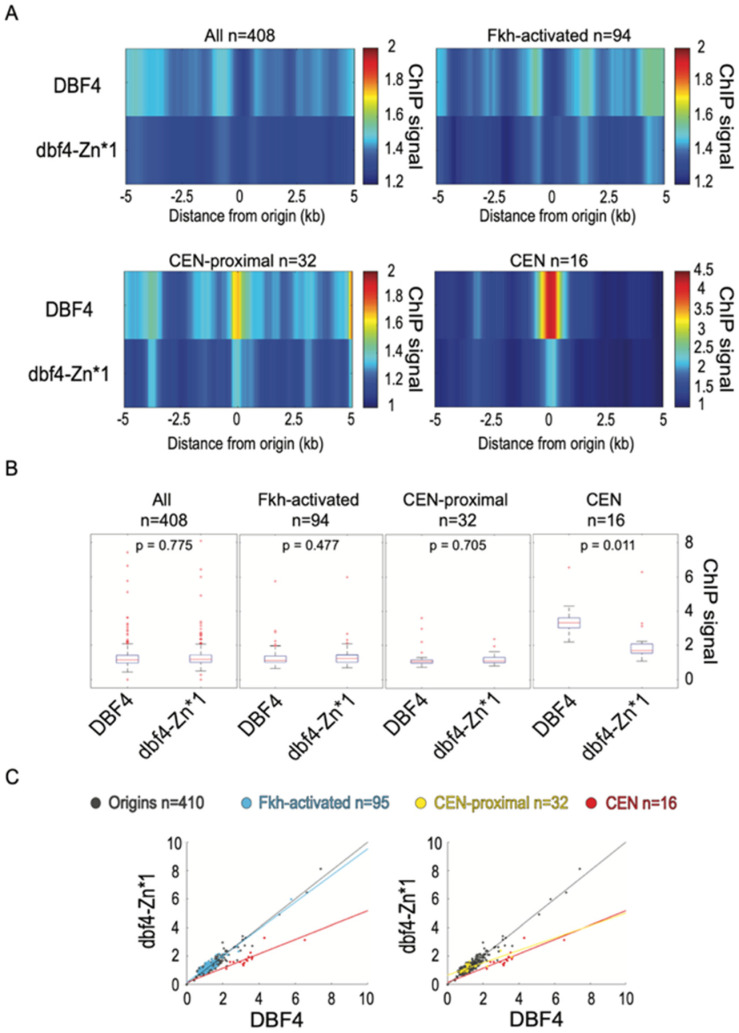
Dbf4-Zn* is defective in its recruitment to CENs. Strains EAy1 (*3xFLAG-DBF4*) and EAy2 (*3xFLAG-dbf4-Zn*1*) were synchronized in G1 phase and subjected to ChIP-seq analysis. (**A**) Heatmaps of averaged ChIP-seq values across 10kbp regions centered on origins (n = 408 because two rDNA origins removed), origin sub-groups, or CENs. (**B**) Boxplot distributions of ChIP-seq values for 500 bp windows centered on the indicated features, subjected to two-sided t-tests. (**C**) Two-dimensional scatter plots of ChIP-seq values for 500 bp windows centered on the indicated features, showing lines of best fit for origin sub-groups and CENs.

**Table 1 genes-13-02202-t001:** Sequences of DNA oligonucleotides used in this study. Oligonucleotides were obtained from International DNA Technologies.

Name	Sequence	Source
∆N-dbf4_F	ATAGGGCGAATTGGAGCTCCACCGCGGTGGCGGCCGCTCTACAAACTTAGACGAACACC	This Study
∆N-dbf4_R	AAAGCTGGGTACCGGGCCCCCCCTCGAGGTCGACGGTATCTCAATACCAGCTTTCTAGC	"
dbf4∆C_F	GACAGCACAGACAGCACAGCCGGTGAAGAAAGAAACGGTAtgaggcgcgccacttctaaa	"
dbf4∆C_R	GATTTTATCACTAAAAGCTACTGCACTTTACGTCGTGTCCcggcgttagtatcgaatcga	"
CTF19∆_F	GTGTGATCTTGTTGATACTAGGTCGGCAAAGAACGCAAATCGATCCCCGGGTTAATTAA	"
CTF19∆_R	GTTTAAGCAAGCCGTCCAGTTGGCAATGCAAATGGAACAGAATTCGAGCTCGTTTAAAC	"
dbf4-Zn-aaHH	CGGTAAAAAATTCCGGATACgcTGAAAATgcTCGTGTAAAATACG	"
dbf4-Zn-CCHc	CATAGTTTCTGAGAAGtgTTTGTCTTTCGCTGAAAACG	"
TB_FLAG-swap	GATGTCATGATCTTTATAATCACCGTCATGGTCTTTGTATCCATTTTCTTCTTTCTTTTCTAAA	"
swap-FLAG-Dbf4	GATTATAAAGATCATGACATCGATTACAAGGATGACGATGACAAGGGTGACGGTGCTGGT TTAAG	"

**Table 2 genes-13-02202-t002:** Genotypes of *S. cerevisiae* strains used in this study. All strains are in the W303 (*RAD5*) background; the parental genotype of SSy161 is shown, whereas for other strains, only differences from this genotype are given.

Name	Genotype	Source
SSy161	MATa ade2-1 ura3-1 his3-11,15 trp1-1 leu2-3,112 can1-100 bar1∆::hisG	Viggiani et al. 2006
SSy162	MATα	"
CVy63	leu2::BrdU-Inc(LEU2)	"
CVy70	MATα ura3::BrdU-Inc(URA3)	"
EAy1	3xFLAG-DBF4 ADE2::FLOPv2x2	This Study
EAy2	3xFLAG-dbf4-Zn*1 (C661A C664A) ADE2::FLOPv2x2	"
HYy143	MATα fkh1Δ::KANMX fkh2-dsm	"
HYy176	dbf4∆C::KANMX leu2::BrdU-Inc(LEU2)	"
HYy207	ctf19Δ::TRP1(Kl) fkh1Δ::KANMX fkh2-dsm ura3::BrdU-Inc(URA3)	"
HYy210	ctf19Δ::TRP1(Kl) ura3::BrdU-Inc(URA3)	"
HYy211	dbf4∆C::HIS3MX ctf19Δ::TRP1 ura3::BrdU-Inc(URA3)	"
HYy215	dbf4∆C::HIS3MX ura3::BrdU-Inc(URA3)	"
HYy217	fkh1Δ::KANMX ura3::BrdU-Inc(URA3)	"
HYy218	fkh1Δ::KANMX fkh2-dsm ura3::BrdU-Inc(URA3)	"
JOSHy1	dbf4∆C::HIS3MX leu2::BrdU-Inc(LEU2)	"
JOSHy2	dbf4∆C::HIS3MX ura3::BrdU-Inc(URA3)	"
MPy35	ADE2::FLOPv2x2	"
MPy74	dbf4-Zn*1 (C661A C664A) leu2::BrdU-Inc(LEU2)	"
MPy76	dbf4-Zn*2 (H680C) leu2::BrdU-Inc(LEU2)	"
MPy86	fkh1Δ::URA3MX dbf4-Zn*1 (C661A C664A) leu2::BrdU-Inc(LEU2)	"
MPy90	fkh1Δ::URA3MX dbf4-Zn*2 (H680C) leu2::BrdU-Inc(LEU2)	"
MPy125	3xFLAG-DBF4 ADE2::FLOPv2x2	"
OAy1123	MATα fkh1Δ::URA3MX fkh2Δ::HIS3MX	"

## Data Availability

Sequencing data is available at GEO (GSE215190).

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
