# Peer review of "Dbf4 Zn-Finger Motif Is Specifically Required for Stimulation of Ctf19-Activated Origins in Saccharomyces cerevisiae"

_genes, 2022, doi:10.3390/genes13122202_

Round 1
Reviewer 1 Report
Eukaryotic DNA replication initiates from many origins with various timing in the S phase. In budding yeast, Fkh1/2 or Ctf19 independently, recruits Dbf4, a regulatory subunit of Cdc7 kinase (or DDK), to early-firing replication origins. DDK phosphorylates the MCM complex to form replicative DNA helicase at origins and thus works as a rate-limiting factor of replication origins. For the recruitment, the C-terminal part of Dbf4 was suggested to work as an interaction domain with Fkh1/2. In this manuscript, the authors constructed the mutations of the Zinc-finger like motif in the C-terminal part of Dbf4 and found them defective in early firing of CEN-proximal origins, which requires Ctf19. The genetic and phenotypic data presented in this manuscript support that the Zinc-finger like motif is required for interaction with Ctf19 but not with Fkh1/2. This is also true for the C-terminal deletion of Dbf4, which diminished the early firing of CEN-proximal origins and weakly reduced the Fkh1/2-dependent early-firing. From these data, the authors further suggest that Fkh1/2 affects DDK by the ways not described before. The authors carefully carried out the experiments and the data presented support their conclusion. However, there is no direct evidence for defective interaction between the mutant Dbf4 and Ctf19 and for reduced recruitment of Dbf4 to CEN-proximal origins. The additional data, that is, the interaction between the mutant Dbf4, Ctf19 and Fkh1/2 proteins and recruitment of Dbf4 to origins will strengthen authors’ conclusion.
Specific comments
1. The data presented in this manuscript indicate some inconsistency with previous report (Fang et al, 2017). The authors have better discuss it more in detail.
2. For the interaction study, mutant proteins must be purified. A mutant protein often has difficulty for purification or for assay by various reasons, for example, by its unstability and/or stickiness. The ChIP assay for Dbf4 recruitment to origins may be hard task (few examples in publications). If the authors have some reasons not to obtain the results from the suggested experiments, they have better describe reasons.
3. The schematic presentation of Dbf4 with a Zinc-finger like motif is helpful for audience.
4. Legend to Figure 3(D) is lacked.
Author Response
We thank the reviewers for their efforts in reviewing this paper. Below, we have attempted to address all comments and criticisms. We have also added some new results (new Fig 5 and text Section 3.6), including ChIP-seq of Dbf4 (which was correctly acknowledged as a likely challenging experiment). We have also added new Western blot analysis of Dbf4, Dbf4∆C, and Dbf4-Zn*1/2 (new Figure S1), providing further evidence that Dbf4∆C is a hypomorph, while the Zn-finger mutations do not appear to alter protein levels. These new results are the work product of two new individuals, whom we have added as co-authors.
- The data presented in this manuscript indicate some inconsistency with previous report (Fang et al, 2017). The authors have better discuss it more in detail.
We think the paper makes very clear that the results with dbf4∆C should not be interpreted for specific targeting defects due to being severe hypomorph. The Abstract and “Perspectives” section makes clear that the mechanism of Fkh1/2 with Dbf4 interaction requires more work.
- For the interaction study, mutant proteins must be purified. A mutant protein often has difficulty for purification or for assay by various reasons, for example, by its unstability and/or stickiness. The ChIP assay for Dbf4 recruitment to origins may be hard task (few examples in publications). If the authors have some reasons not to obtain the results from the suggested experiments, they have better describe reasons.
We performed ChIP-seq of Dbf4 and Dbf4-Zn*1. The results directly support our conclusion of a recruitment defect for stimulation of CEN-proximal origins. However, we do not detect Dbf4 at other origins with sufficient signal to make conclusion about recruitment to other origins. However, the QBU data are clear about function of Dbf4-Zn* in stimulation of Fkh-activated origins. These new results are included as Fig. 5 and text Section 3.6.
- The schematic presentation of Dbf4 with a Zinc-finger like motif is helpful for audience.
We added this as Fig. 1A.
- Legend to Figure 3(D) is lacked.
Thanks, this has been corrected.
Thanks again.
Reviewer 2 Report
Execution of the DNA replication is essential for maintaining genome stability. One of the key regulatory points for DNA replication is the initiation step, where DDK is required to activate the MCM2-7 helicase. Petrie, et al., use budding yeast as model to understand how DDK is recruited to specific replication origins to promote origin firing. Previous work (largely from the Aparicio lab) has defined Fkh1/2 as a critical factor needed for the activation of early-firing replication origins in budding yeast. While Fkh1/2 are necessary for a subset of origins, Ctf19 is needed CEN-proximal origins. How these factors compete for DDK is an open and interesting question.
Through careful genetic analysis, the authors have found that the C-terminal region of Dbf4 is critical for its ability to promote replication initiation. Furthermore, the C-terminal Zn finger in Dbf4 appears to be critical for recruiting DDK to Ctf19-depenendent replication origins, but has much less of an effect on Fhh-dependent origins. Therefore, the authors have provided a concrete example of how origins use different recruitment mechanisms to access limiting replication factors. This manuscript is very well written, experiments are well controlled and makes an important contribution the replication field. I have only minor comments listed below.
Ln 158-160: This claim is hard to see from the data provided. Are we supposed to compare the top plates (at 37) to the bottom plates at 37? Maybe it would be easier to see if they were side by side on the same plate and/or quantified in some way. I only see that the colonies may look smaller in the double mutants. It’s hard to tell if there is actually a reduction in CFUs between the single and double mutants without quantification.
Ln 217-219: While the dbf4-Zn mutant clearly has a preferential effect on Ctf19 origins, the data in 3B and 3C indicate that Fkh-activated origins are reduced in the dbf4-Zn mutants. I would suggest that authors tone down the language to account for this.
Ln 43: … early0figine origins in the S. cerevisiae genome…
Ln71: bearing mutations
Ln 123: define QBU when it first appears
Ln126: Need GEO accession #
Ln136: with a focus
Ln149: do not phenocopy
Ln150: at the permissive temperature
Ln183: as inferred below (at least on my version)
Figure 3: The authors may want to change their color scheme on 3B/D. I’m not RG color blind, but I still had a hard time seeing the color distribution.
*There is no legend for 3D
Ln255: (Fig. 3A-C) – I would suggest directing readers to the 30’ time point where this is most evident
Ln309 (and throughout the manuscript): It seems the authors are interpreting their genetic data to infer a physical interaction between Fkh and Dbf4. Given that no evidence of a direct physical association is presented in this work, I would suggest the authors tone down the language.
Author Response
We thank the reviewers for their efforts in reviewing this paper. Below, we have attempted to address all comments and criticisms. We have also added some new results (new Fig 5 and text Section 3.6), including ChIP-seq of Dbf4 (which was correctly acknowledged as a likely challenging experiment). We have also added new Western blot analysis of Dbf4, Dbf4∆C, and Dbf4-Zn*1/2 (new Figure S1), providing further evidence that Dbf4∆C is a hypomorph, while the Zn-finger mutations do not appear to alter protein levels. These new results are the work product of two new individuals, whom we have added as co-authors.
Ln 158-160: This claim is hard to see from the data provided. Are we supposed to compare the top plates (at 37) to the bottom plates at 37? Maybe it would be easier to see if they were side by side on the same plate and/or quantified in some way. I only see that the colonies may look smaller in the double mutants. It’s hard to tell if there is actually a reduction in CFUs between the single and double mutants without quantification.
We agree. The colonies are smaller indicative of slower growth; we are not claiming a difference in CFUs.
Ln 217-219: While the dbf4-Zn mutant clearly has a preferential effect on Ctf19 origins, the data in 3B and 3C indicate that Fkh-activated origins are reduced in the dbf4-Zn mutants. I would suggest that authors tone down the language to account for this.
We think the language is appropriately qualified. For example, we stated: “The results are consistent with…” and “The results further suggest…” previous lines 218 and 219 (now lines 246 and 247)
Ln 43: … early0figine origins in the S. cerevisiae genome…
Corrected
Ln71: bearing mutations
Change accepted
Ln 123: define QBU when it first appears
Corrected
Ln126: Need GEO accession #
Added
Ln136: with a focus
Change accepted
Ln149: do not phenocopy
Correct as is
Ln150: at the permissive temperature
Change accepted
Ln183: as inferred below (at least on my version)
Text has been edited to clarify that reference was to the earlier growth plating data (Fig. 1)
Figure 3: The authors may want to change their color scheme on 3B/D. I’m not RG color blind, but I still had a hard time seeing the color distribution.
Done throughout
*There is no legend for 3D
Corrected.
Ln255: (Fig. 3A-C) – I would suggest directing readers to the 30’ time point where this is most evident
Line 255 doesn’t refer to the time-course. The reviewer must be referring to Fig 2 but it’s unclear which comparison.
Ln309 (and throughout the manuscript): It seems the authors are interpreting their genetic data to infer a physical interaction between Fkh and Dbf4. Given that no evidence of a direct physical association is presented in this work, I would suggest the authors tone down the language.
We have changed the term recruit in a few cases to functional targeting or stimulation of origin firing. Still, we have now provided direct data for the role of the Zn-finger in CEN-recruitment in new section 3.6 and directly address the limitation of the conclusion of direct recruitment.
Thanks again.
Round 2
Reviewer 1 Report
The authors improved their manuscript well based on my comments. I thus think this form is suitable for publication.